

# Inbreeding depression in one of the last DFTD-free wild populations of Tasmanian devils

Rebecca M. Gooley[1], Carolyn J. Hogg[1], Samantha Fox[2,3], David Pemberton[2], Katherine Belov[1] and Catherine E. Grueber[1,4]

[1] School of Life and Environmental Sciences, University of Sydney, Sydney, New South Wales, Australia
[2] Save the Tasmanian Devil Program, Hobart, Tasmania, Australia
[3] Toledo Zoo, Toledo, OH, United States of America
[4] San Diego Zoo Global, San Diego, CA, United States of America

## ABSTRACT

**Background**. Vulnerable species experiencing inbreeding depression are prone to localised extinctions because of their reduced fitness. For Tasmanian devils, the rapid spread of devil facial tumour disease (DFTD) has led to population declines and fragmentation across the species' range. Here we show that one of the few remaining DFTD-free populations of Tasmanian devils is experiencing inbreeding depression. Moreover, this population has experienced a significant reduction in reproductive success over recent years.

**Methods**. We used 32 microsatellite loci to examine changes in genetic diversity and inbreeding in the wild population at Woolnorth, alongside field data on breeding success from females to test for inbreeding depression.

**Results**. We found that maternal internal relatedness has a negative impact on litter sizes. The results of this study imply that this population may be entering an extinction vortex and that to protect the population genetic rescue should be considered. This study provides conservation managers with useful information for managing wild devils and provides support for the "Wild Devil Recovery Program", which is currently augmenting small, isolated populations.

## INTRODUCTION

For threatened species, a reduction in reproductive success can severely impact population persistence. The Tasmanian devil, *Sarcophilus harrisii,* is one such species that has declined up to 80% in areas affected by an infectious clonal cancer, devil facial tumour disease (DFTD) (*Loh et al., 2006*; *Pye et al., 2016*; *Lazenby et al., 2018*). As the apex carnivore in Tasmania, devil population declines are causing trophic cascades in the Tasmanian ecosystem (*Hollings et al., 2014*) and recent modelling has indicated that these populations will begin to succumb to small-population genetic pressures (*Grueber et al., 2019*). Declining populations are at risk of reduced gene flow and loss of genetic diversity (relative to larger, more connected populations) as an outcome of genetic drift and inbreeding (*Charlesworth & Willis, 2009*).

Corresponding author
Catherine E. Grueber,
catherine.grueber@sydney.edu.au

Since the discovery of DFTD in the mid-1990s in the north-east of Tasmania, the national and international conservation community has come together and research into Tasmanian devil biology has grown rapidly, including studies of DFTD epidemiology (e.g., *Hamede, McCallum & Jones, 2008*; *McCallum et al., 2009*; *Hamede et al., 2012*), devil behaviour (e.g., *Sinn et al., 2014*), ecological impacts (e.g., *Hollings et al., 2014*), population genetics (e.g., *Lachish et al., 2011*; *Grueber et al., 2015*; *Epstein et al., 2016*; *Hendricks et al., 2017*), *ex situ* conservation (e.g., *Hogg et al., 2016*) and translocations (e.g., *Rogers et al., 2016*; *Thalmann et al., 2016*; *Wise et al., 2016*; *Grueber et al., 2017*). As DFTD spread from the north-east across Tasmania, devil populations have been monitored by the Save the Tasmanian Devil Program (STDP) since 2004 (*Lazenby et al., 2018*). The disease has spread in a generally south-westward direction, and is now known to exist across most of the state of Tasmania, with disease-free areas limited to the north-west and south-west of the state (*Pemberton, 2019*). One of these last-known DFTD-free populations is located at Woolnorth (40.77°S, 144.77°E), north-west Tasmania (*Farquharson et al., 2018*; *Lazenby et al., 2018*). Since 2014, this population has suffered an extreme decline in reproductive output, the cause of which remains unclear (*Farquharson et al., 2018*). That is, between 2004 and 2009, the proportion of females breeding at Woolnorth was between 60 and 80%, however between 2014 and 2016 the proportion of females breeding was approximately 20%, a 40–60% reduction in a five-year period (*Farquharson et al., 2018*). Although for a number of carnivorous marsupials a correlation between climate and litter sizes has been shown (*Fisher, Owens & Johnson, 2001*; *Collett, Baker & Fisher, 2018*), this does not appear to be the sole driver of the reduction of female reproductive output in Tasmanian devils at Woolnorth (*Farquharson et al., 2018*).

Here we aimed to test whether the observed decline in wild devil reproductive fitness (specifically litter size; devils have a maximum litter size of four pouch young; *Guiler, 1970*) is a result of accumulating inbreeding. Inbreeding depression occurs when an accumulation of deleterious recessive alleles lowers individual heterozygosity, negatively impacting individual fitness relative to less-inbred individuals or populations (*Keller & Waller, 2002*; *Frankham et al., 2017*). Previous genetic research on a captive Tasmanian devil population revealed inter-individual variation in inbreeding, but no signs of inbreeding depression (*Gooley et al., 2017*). Although inbreeding depression is easier to study in controlled environments (such as captivity), it may be more consequential in the wild, as environmental conditions are more severe (*Joron & Brakefield, 2003*; *Armbruster & Reed, 2005*; *De Boer et al., 2015*). Thus, studies of inbreeding depression in captive environments may underestimate the impact of inbreeding on fitness in the wild (*Kristensen, Loeschcke & Hoffmann, 2008*; *Gooley et al., 2017*). In addition, wild populations that experience inbreeding depression are more vulnerable to extinction (*Keller & Waller, 2002*), and so isolated populations may need genetic rescue to combat the effects of inbreeding (*Frankham, 2015*; *Frankham et al., 2017*).

Here we use multilocus heterozygosity to investigate inbreeding and inbreeding depression in the DFTD-free population of devils at Woolnorth. We aimed to test: (1) whether inbreeding is occurring in the devil population at Woolnorth, and (2) whether inbreeding is associated with the observed reduction in reproduction (specifically litter

sizes). The results of this study will inform the ongoing management of fragmented devil populations in the face of DFTD.

## MATERIALS & METHODS

### Sample collection and genotyping

Samples were collected by the STDP following their Standard Operating Procedure (see Appendix 5 in *Hogg et al., 2019*) and shared with the University of Sydney for genetic analysis. DNA samples and corresponding reproductive and demographic data were available for years 2006, 2007, 2009, 2014, 2015 and 2016. Reproductive output for females was taken as the estimated count of offspring produced (i.e., "litter size"), following *Farquharson et al. (2018)*. Female devils are limited to a maximum of 4 offspring per breeding event (*Guiler, 1970*). As is standard practice for documenting reproductive output in Tasmanian devils (following *Keeley et al., 2012*; *Farquharson et al., 2018*), litter size was estimated by the presence and count of pouch young for all years except 2009. The 2009 monitoring trip occurred later in the year, so litter size was estimated by the presence and count of active teats (indicating pouch young had been denned). As devils are marsupials, pouch young attach to the teat shortly after birth, and remain attached for approximately 4 months. Unoccupied teats where no pouch young attach after birth will noticeably regress (*Hesterman, Jones & Schwarzenberger, 2008*). Denned devils (∼5–10 months post birth) will continue to suckle keeping the teat active providing an indication of the number of offspring that had birthed and attached to a teat. In total, 168 wild Tasmanian devils (90 females and 78 males) were included in this study; none provided replicate measurements. Male reproductive output could not be examined in this study due to the open nature of the population, making pedigree reconstruction from genetic data difficult.

DNA from ear biopsy samples from the 2006, 2007 and 2009 monitoring trips had been previously extracted (*Hendricks et al., 2017*), whilst samples from 2014, 2015 and 2016 were extracted using a phenol-chloroform technique (*Sambrook, Fritsch & Maniatis, 1989*) and stored at −20 °C. Samples were genotyped with 32 putatively neutral microsatellite markers following *Gooley et al. (2017)* and *Jones et al. (2003)*. A randomly chosen set of 7% were re-genotyped to estimate genotyping error. We tested for null alleles at each locus using Micro-Checker (*Van Oosterhout et al., 2004*), null allele frequencies per year and per locus were calculated using the method of *Brookfield (1996)* and tabulated via Genepop (*Raymond & Rousset, 1995*; *Rousset, 2008*). GenAlEx (*Peakall & Smouse, 2006*; *Peakall & Smouse, 2012*) was used to calculate observed ($H_O$) and expected heterozygosity ($H_E$) for each locus, each year, and conduct Hardy-Weinberg exact tests.

### Inbreeding and inbreeding depression

Internal relatedness (IR), a multilocus heterozygosity statistic that is expected to be positively correlated with individual inbreeding coefficient (*Amos et al., 2001*), was calculated using the function GENHET (*Coulon, 2010*) for R (*R Core Team, 2019*). IR incorporates allele frequencies, because there is a higher chance that rare-allele homozygosity is the result of inbred mating, relative to common-allele homozygosity

(*Amos et al., 2001*). All available samples, male and female, were used to estimate allele frequencies and calculate IR, to minimise impact of yearly allele frequency changes on calculated IR values. Across our dataset, IR was very highly correlated with other common measures of multilocus heterozygosity (such as standardised observed heterozygosity, and heterozygosity-by-loci; all absolute correlation coefficients were ≥0.94), so we focussed our main statistical analyses on IR.

We examined whether inbreeding was accumulating among individuals in the population by testing for a change in IR over time using a linear model fitted in R with year as the fixed predictor and IR as the response ($N = 168$). We evaluated change in the population-level of inbreeding ($F_{IS}$), calculated using the package *hierfstat* (*Goudet, 2005*) for R.

To interpret associations between heterozygosity and litter sizes as inbreeding depression, molecular data must reflect variation in inbreeding levels among individuals, i.e., identity disequilibrium (*Szulkin, Bierne & David, 2010*). This variation was quantified with the $g_2$ statistic (*David et al., 2007*; *Szulkin, Bierne & David, 2010*), using the package *inbreedR* (*Stoffel et al., 2016*) for R, with its precision evaluated using 1,000 Monte Carlo iterations.

We tested for inbreeding depression by determining whether IR was a predictor of female reproduction using linear regression. The equations used for the regression were:

- Litter size: $logit\left(\frac{littersize}{4}\right) = \beta_0 + \beta_1 IR_i + \beta_2 age_i + \beta_3 year_i + \varepsilon_i$
- Probability of breeding: $logit\left(P_{breed}\right) = \beta_0 + \beta_1 IR_i + \beta_2 age_i + \beta_3 year_i + \varepsilon_i$

where $\beta_0$ is the intercept, $\beta_{1-3}$ are regression coefficients associated with the specified predictor variables, and $\varepsilon_i$ is the error term. We also attempted to add year as a random effect (e.g., following *Barr et al., 2013*), but only our litter size model converged. Those results were qualitatively similar to our main findings, and so are presented in Supplementary Results for comparison. Litter size was modelled as a binomial response using the cbind function in the R *base* package, where the number of offspring was the number of binomial "events" and the number of trials was 4 (maximum possible litter size). The litter size model was fitted twice: with the litter sizes of all females ($N = 90$), and with only those females that showed evidence of breeding (i.e., producing 1 or more offspring, $N = 36$). Inbreeding depression is expected to produce a negative slope for IR (our predictor of interest), i.e., increased IR is associated with decreased litter sizes. Age (based on tooth wear observations, *Pemberton, 1990*) and year were also included as continuous fixed predictors (with year $= 0$ for 2006). Model selection was conducted using an information theoretic approach following *Grueber et al. (2011)*, with standardisation following Gelman (*Gelman, 2008*) using the package *arm* (*Gelman & Su, 2015*), and multimodel inference performed using the package *MuMIn* (*Bartoń, 2009*). We report the final model effect sizes and their 95% confidence intervals (based on 1.96 x adjusted SE), in addition to their relative importance (RI, sum of Akaike weights), and the $R^2$ of the global model calculated using the package *rsq* (*Zhang, 2018*).

To consider the effects of individual loci in generating heterozygosity-fitness correlations (i.e., "local effects"; see *Szulkin, Bierne & David, 2010*), we tested the hypothesis that heterozygosity of some loci may be individually more informative of variation in fitness

than a statistic that measures the combined effect of all loci. This prediction would be upheld if an overall effect of multilocus heterozygosity is driven by only one or a few loci. It is important to note that, under inbreeding, heterozygosity values of individual loci are not independent, and so it is the magnitudes of relative effect sizes that are important (*Szulkin, Bierne & David, 2010*). The most widely-cited method for testing the local effects hypothesis uses multiple regression, whereby each locus is fitted simultaneously, and the slopes compared (*Szulkin, Bierne & David, 2010*). This method requires a complete dataset (no missing genotype data, as incomplete cases are excluded from standard multiple regressions; *Nakagawa & Freckleton, 2008*) and, to avoid overfitting, a large sample size relative to the number of loci. In linear regression, a ratio of cases:predictors of approximately 10–20 is recommended for fitting a statistically robust regression (*Harrell, 2015*). For our dataset, we had 32 loci and 69 complete cases (females with no missing genotype data), falling far short of the recommended data required for this method. We therefore approached the need to model our loci separately but simultaneously, and alongside multilocus heterozygosity, by using an information theoretic approach. We considered the effect of a locus' heterozygosity on fitness as an independent local-effect hypothesis by fitting separate models for each locus, and considered all loci collectively by ranking and comparing their AIC values to draw inference based on both the degree of support for each of model, and the effect sizes (slopes) of heterozygosity. Our model set included:

- Single-locus models, which include observed heterozygosity of each locus fitted separately (coded as a 0/1 for homozygote/heterozygote; following *Grueber, Wallis & Jamieson (2013)*; 32 models in total. These models all also include informative non-genetic parameters (as in the "base" model, see below).
- A "base" model, which excluded heterozygosity data altogether. This model can be considered as our null hypothesis for the purposes of examining local effects, including any non-genetic parameters that were found to influence fitness in our main analysis (namely year, see Results).
- An "$H_O$" multilocus heterozygosity model, which fits observed $H_O$ averaged across all loci. The $H_O$ statistic was used as our multilocus measure (as opposed to IR) to facilitate direct comparison with the single-locus models. The $H_O$ model also includes informative non-genetic parameters (as in the "base" model).

All 34 models were quantitatively ranked and compared using AIC; our "base" and "$H_O$" models served as reference points for calculating $\Delta$AIC. Models with lower AIC are interpreted as having stronger support, and $|\Delta AIC|<2$ as models with similar levels of support (*Burnham & Anderson, 1998*). We note that the reduced dataset of $N = 69$ females produced qualitatively the same results as in our main analysis.

## RESULTS

Inferred null allele frequencies were very low for most loci/years (Table 1), and we had little missing data: >90% of individuals were successfully genotyped for >90% of loci. Genotyping error rate was 0.6%. Microsatellite diversity of Woolnorth devils was low (Table 1), and

similar to observations of other wild sites and captive populations (e.g., *Gooley et al., 2017*; *Storfer et al., 2017*; *Grueber et al., 2019*). Levels of IR remained constant across the study period (linear regression: $\beta_{year} = 0.003 \pm 0.005$ SE, $p = 0.546$; $\beta_0 = -5.621 \pm 9.295$ SE, $p = 0.546$, $N = 168$ devils, Fig. 1A). The same result was obtained when using observed heterozygosity, which does not take allele frequencies into account (linear regression: $\beta_{year} = -0.002 \pm 0.002$ SE, $p = 0.303$; $\beta_0 = 4.400 \pm 3.896$ SE, $p = 0.260$, $N = 168$ devils). Similarly, considering inbreeding at the population level in respect of Hardy–Weinberg equilibrium ($F_{IS}$), we also found no trend over time (Fig. 1B).

We were able to assess inbreeding using our dataset as we detected statistically significant identity disequilibrium ($g_2 = 0.017$, SE $= 0.007$, $p$-value $= 0.003$), indicating that variation at our molecular markers reflects variation in the level of inbreeding among individuals. We found evidence that inbreeding depression is occurring in the female devil population at Woolnorth as IR had a strong negative effect on overall female litter sizes (increased homozygosity [IR] was associated with decreased fitness) (Table 2). We found little evidence of an effect of IR on propensity to breed at all (weak effect size, wide error, poor relative importance; Table 2), but when examining only those females that had at least one offspring, the effect of IR that was seen for overall litter size was confirmed (Table 2). We therefore infer our overall results are not driven by effects of IR on breeding *per se*, but that the inbreeding depression applies primarily to litter size specifically.

Considering locus-by-locus effects of heterozygosity on litter size, we found compelling evidence that three loci (Sha3o, Sha32 and Sha013) are stronger determinants of litter size than multilocus heterozygosity. This result is inferred based on those single-locus models having substantially greater support than that of the multilocus estimator ($\Delta$AIC >4; Table 3). However, the relative effect sizes of these loci (slope of heterozygosity) are all weaker than the multilocus model (Table 3) suggesting that these findings are not consistent with strong local effects. For example, two of the strongest-effect loci (Sh3o and Sha32) showed reduced fitness in heterozygotes relative to homozygotes; i.e., a negative effect of heterozygosity, which is opposite to predictions under inbreeding depression and opposite to the main effect of $H_O$ (Table 3). Although Sha013 showed improved fitness in heterozygotes (consistent with predictions), its effect was much weaker than seen in the multilocus model ($\beta_{H_O}$(Sha013) $= 1.196 \pm 0.399$ SE, while $\beta_{H_O}$(multilocus) $= 4.260 \pm 1.847$ SE; Table 3). For the three loci with greatest evidence of an effect of an effect of heterozygosity on fitness, Sh3o and Sha013 had moderate rates of heterozygosity, while for Sha32 only five heterozygotes were observed in the reduced sample (frequency 0.072, $N = 69$; Table 3). Of these five Sha32 heterozygotes, four were observed in the "early" part of the study, when reproductive rates were generally high (negative effect of Year in our modelling, Tables 2, 3), but only two produced litters, which were small (two joeys each). The observed Sha32 data for this small sample set is therefore consistent with the negative trend seen in the modelling results (heterozygotes produced fewer offspring than expected); more data would be required to confirm this pattern.

Five further loci (Sha040, Sha039, Sh2g, Sh2p and Sh6e) have similar levels of single-locus model support as the multilocus estimator ($|\Delta$AIC$| < 2$); their effects on fitness were all positive (in line with predictions and consistent with the multilocus predictor),

**Table 1  Genetic variation of 32 polymorphic microsatellite loci in the Woolnorth Tasmanian devil population.**  Diversity is measured by number of alleles (Na), observed heterozygosity ($H_O$), unbiased estimate of expected heterozygosity ($H_E$) and Hardy-Weinberg Exact test ($p$-value). Total number of devils $N = 168$.

| Locus[2] | N | Na | $H_O$ | $H_E$ | $p$-value | Estimated null allele frequencies per year[1] | | | | | |
|---|---|---|---|---|---|---|---|---|---|---|---|
| | | | | | | 2006 | 2007 | 2009 | 2014 | 2015 | 2016 |
| Sh2b | 147 | 2 | 0.340 | 0.378 | 0.239 | 0.000 | 0.010 | 0.015 | 0.692 | 0.333 | 0.008 |
| Sh2g | 167 | 3 | 0.701 | 0.646 | 0.053 | 0.000 | 0.000 | 0.000 | 0.000 | 0.000 | 0.100 |
| Sh2i | 168 | 3 | 0.411 | 0.406 | 0.443 | 0.000 | 0.000 | 0.000 | 0.000 | 0.000 | 0.043 |
| Sh2p | 168 | 3 | 0.667 | 0.617 | 0.300 | 0.000 | 0.000 | 0.000 | 0.046 | 0.061 | 0.042 |
| Sh2v | 168 | 6 | 0.548 | 0.587 | 0.738 | 0.069 | 0.022 | 0.000 | 0.000 | 0.000 | 0.030 |
| Sh3a | 155 | 3 | 0.226 | 0.245 | 0.078 | NA | 0.000 | 0.000 | 0.551 | 0.230 | 0.026 |
| Sh3o | 168 | 4 | 0.464 | 0.522 | 0.129 | 0.027 | 0.000 | 0.054 | 0.000 | 0.006 | 0.082 |
| Sh5c | 160 | 3 | 0.069 | 0.067 | 0.977 | 0.000 | NA | 0.000 | 0.468 | 0.186 | 0.000 |
| Sh6e | 168 | 2 | 0.435 | 0.412 | 0.452 | 0.629 | 0.404 | 0.558 | 0.415 | 0.607 | 0.555 |
| Sh6L | 167 | 2 | 0.138 | 0.139 | 0.943 | 0.000 | 0.000 | 0.056 | 0.156 | 0.000 | 0.000 |
| Sha001 | 164 | 3 | 0.085 | 0.083 | 0.955 | 0.577 | 0.264 | NA | 0.000 | 0.000 | 0.000 |
| Sha008 | 161 | 3 | 0.547 | 0.534 | 0.769 | 0.077 | 0.003 | 0.097 | 0.265 | 0.179 | 0.000 |
| Sha009 | 163 | 4 | 0.319 | 0.297 | 0.954 | 0.516 | 0.199 | 0.000 | 0.145 | 0.019 | 0.000 |
| Sha010 | 161 | 7 | 0.826 | 0.778 | 0.757 | 0.163 | 0.000 | 0.000 | 0.230 | 0.134 | 0.000 |
| Sha011 | 167 | 2 | 0.329 | 0.386 | 0.061 | 0.000 | 0.010 | 0.024 | 0.000 | 0.126 | 0.114 |
| Sha012 | 156 | 3 | 0.487 | 0.538 | 0.000 | 0.000 | 0.016 | 0.191 | 0.307 | 0.331 | 0.024 |
| Sha013 | 162 | 7 | 0.710 | 0.675 | 0.718 | 0.000 | 0.058 | 0.000 | 0.200 | 0.164 | 0.000 |
| Sha014 | 165 | 4 | 0.491 | 0.525 | 0.108 | 0.000 | 0.073 | 0.000 | 0.217 | 0.140 | 0.000 |
| Sha015 | 155 | 2 | 0.471 | 0.471 | 0.978 | 0.000 | 0.114 | 0.000 | 0.542 | 0.176 | 0.000 |
| Sha023 | 156 | 5 | 0.436 | 0.423 | 0.998 | 0.000 | 0.036 | 0.017 | 0.518 | 0.096 | 0.000 |
| Sha024 | 148 | 2 | 0.209 | 0.199 | 0.486 | 0.000 | 0.000 | 0.173 | 0.603 | 0.286 | 0.160 |
| Sha025 | 166 | 2 | 0.193 | 0.231 | 0.037 | 0.217 | 0.107 | 0.000 | 0.139 | 0.194 | 0.006 |
| Sha026 | 164 | 3 | 0.226 | 0.233 | 0.667 | NA | 0.026 | 0.000 | 0.261 | 0.179 | 0.004 |
| Sha028 | 148 | 5 | 0.264 | 0.241 | 0.970 | 0.000 | 0.000 | 0.000 | 0.647 | 0.374 | 0.000 |
| Sha032 | 147 | 3 | 0.061 | 0.060 | 0.986 | NA | 0.000 | 0.000 | 0.707 | 0.419 | NA |
| Sha033 | 166 | 2 | 0.331 | 0.301 | 0.178 | 0.000 | 0.104 | 0.000 | 0.121 | 0.049 | 0.003 |
| Sha034 | 166 | 3 | 0.193 | 0.200 | 0.580 | NA | 0.000 | 0.021 | 0.175 | 0.000 | 0.117 |
| Sha036 | 165 | 2 | 0.248 | 0.295 | 0.048 | 0.000 | 0.000 | 0.024 | 0.261 | 0.094 | 0.122 |
| Sha037 | 164 | 6 | 0.610 | 0.688 | 0.000 | 0.000 | 0.045 | 0.067 | 0.245 | 0.026 | 0.034 |
| Sha039 | 160 | 4 | 0.400 | 0.407 | 0.961 | 0.000 | 0.026 | 0.118 | 0.252 | 0.221 | 0.082 |
| Sha040 | 165 | 5 | 0.612 | 0.599 | 0.000 | 0.000 | 0.000 | 0.171 | 0.048 | 0.124 | 0.019 |
| Sha042 | 163 | 2 | 0.313 | 0.297 | 0.479 | 0.000 | 0.149 | 0.033 | 0.217 | 0.122 | 0.095 |

**Notes.**
[1] Estimated using the method of *Brookfield (1996)*; NA indicates a locus that was monomorphic in the specified year dataset.
[2] The ten "Sh" markers were developed by *Jones et al. (2003)*; the remaining 22 "Sha" markers were developed by *Gooley et al. (2017)*. *Jones et al. (2003)*.

and all were weaker than the multilocus predictor (compare the $\beta_{H_O}$ values in Table 3). No other single-locus models were superior to the multilocus model for explaining litter size (Table 3). As none of the slopes of our strongest single-locus effects are of greater

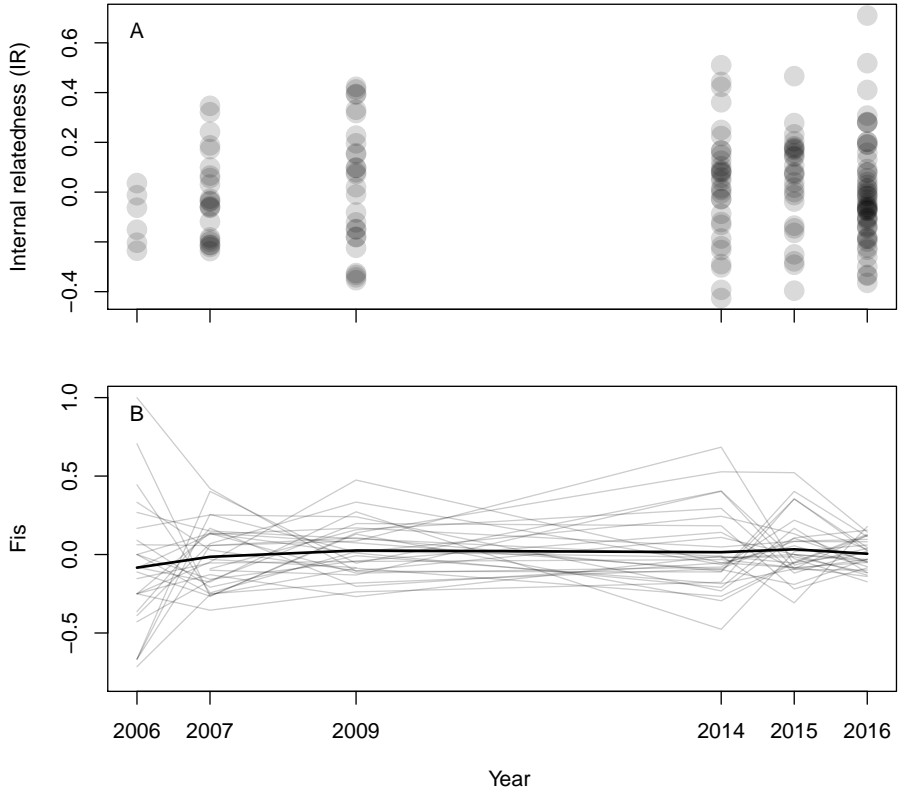

**Figure 1** **Inbreeding in wild Tasmanian devils at Woolnorth (males and females) over time.** (A) shows individual-level inbreeding recorded as internal relatedness (IR); each point is an individual devil. (B) shows population-level inbreeding (deviation from Hardy-Weinberg expectations, $F_{IS}$); each of the faint grey lines is a microsatellite locus, with the heavy black line illustrating the mean trend. Note: annual monitoring trips were not conducted in 2008, 2010, 2011, 2012 nor 2013.

magnitude than the main effect of multilocus heterozygosity, we interpret our results as consistent with general (genome-wide) effects, i.e., inbreeding depression.

## DISCUSSION

Here, we show that one of the last-known DFTD-free wild populations of Tasmanian devils is experiencing inbreeding depression. Although our data did not detect an increase in inbreeding over the timescale of our study, we did show that maternal IR has a negative impact on reproductive output (litter size) in wild devils. A previous study observed a significant decline in reproduction over time in this population (*Farquharson et al., 2018*). It is unclear whether inbreeding depression may be either partially responsible for this trend, or a worrying consequence of it. However when these past observations are considered alongside the findings of the current study, we suggest that the Woolnorth population may be close to a tipping point, whereby inbreeding reduces reproductive rates (perhaps in concert with other factors), which in turn further reduces population size and exacerbates the occurrence of inbreeding and inbreeding depression. This raises the

**Table 2 Predictors of reproductive success in female Tasmanian devils.** Predictors have been standardised, and are the conditional average results derived from an information theoretic model selection process (see Methods); full model sets are provided in Table S1.

| Model | N | Predictor | Estimate | SE | RI | $R^{2a}$ |
|---|---|---|---|---|---|---|
| Litter size | 90 | Intercept | −1.073 | 0.146 | | 0.371 |
| | | Age | 0.296 | 0.297 | 0.36 | |
| | | IR | −0.953 | 0.275 | 1.00 | |
| | | Year | −2.367 | 0.285 | 1.00 | |
| Breeding | 90 | Intercept | −0.489 | 0.262 | | 0.311 |
| | | Age | 0.605 | 0.556 | 0.38 | |
| | | IR | −0.737 | 0.521 | 0.49 | |
| | | Year | −2.527 | 0.561 | 1.00 | |
| Litter size 1+ | 36 | Intercept | 1.304 | 0.228 | | 0.312 |
| | | Age | −0.896 | 0.445 | 0.77 | |
| | | IR | −1.047 | 0.439 | 1.00 | |
| | | Year | −0.767 | 0.414 | 0.66 | |

**Notes.**
N, sample size; SE, adjusted standard error; IR, internal relatedness; RI, relative importance (sum of Akaike weights).
[a]$R^2$ is derived from the global model.

management option of genetic rescue for Woolnorth, whereby supplementation could increase the reproductive fitness of this population, which is now effectively isolated due to devil facial tumour disease causing 80% declines in adjacent devil populations (*Whiteley et al., 2015*; *Lazenby et al., 2018*).

Small populations that exist in fragmented landscapes are expected to increase in mean inbreeding levels over time (*Wright, Tregenza & Hosken, 2007*; *Frankham et al., 2017*) and monitoring this process is an important element of genetic management in conservation (*Fredrickson et al., 2007*; *La Haye et al., 2012*). Interestingly, for our study, the effects of inbreeding were most influential on litter size and not on a female's propensity to breed. This result suggests inbreeding as a likely cause of the decline in litter size previously reported (*Farquharson et al., 2018*). Given the short time-frame of the study (2006–2016), our failure to detect a corresponding change in IR over time may indicate that a substantive increase in population mean inbreeding levels is yet to occur. This interpretation is not unprecedented: for example, the southernmost Swedish population of arctic fox did not show an increase in inbreeding coefficients until four years after population fragmentation that occurred in the late 1990s (*Noren et al., 2016*). In any case, the declining reproductive output seen here, and previously (*Farquharson et al., 2018*), could lead to a decrease in effective population size. As of 2018, the low reproductive output of the Woolnorth population continues (STDP, *unpublished data*). As a short-lived carnivorous marsupial species, ongoing reductions in litter sizes will likely impact long-term population dynamics (*Fisher, Owens & Johnson, 2001*). If this is an accurate interpretation, the likely consequence of these processes will be an eventual increase in inbreeding, and a strengthening of its negative effects. To test this hypothesis, it will be important to continue monitoring the trajectory of demographic and genetic processes in this population, given its importance as the last DFTD-free wild population of Tasmanian devils.

**Table 3** **Locus-by-locus effects of heterozygosity on litter size of** $N = 69$ **female devils at Woolnorth with complete genotyping data.** Models with "Locus" IDs include a 0/1 predictor for individual heterozygosity at that locus; the "$H_O$" model uses multilocus observed heterozygosity, while the "Base" model excludes heterozygosity data altogether.

| Locus | AIC | $\Delta AIC_{base}$[1] | $\Delta AIC_{H_O}$[2] | $\beta_0$ | $SE(\beta_0)$ | $\beta_{year}$ | $SE(\beta_{year})$ | $\beta_{H_O}$ | $SE(\beta_{H_O})$ | $H_O$[3] |
|---|---|---|---|---|---|---|---|---|---|---|
| **Sh3o** | **176.9** | **−9.0** | **−5.5** | **1.490** | **0.300** | **−0.362** | **0.044** | **−1.178** | **0.370** | **0.377** |
| **Sha32** | **177.8** | **−8.1** | **−4.6** | **1.422** | **0.290** | **−0.388** | **0.045** | **−1.810** | **0.623** | **0.072** |
| **Sha013** | **178.1** | **−7.8** | **−4.3** | **0.360** | **0.351** | **−0.387** | **0.046** | **1.196** | **0.399** | **0.739** |
| Sha040 | 181.5 | −4.4 | −0.9 | 0.669 | 0.305 | −0.366 | 0.044 | 0.833 | 0.336 | 0.522 |
| Sha039 | 181.9 | −4.0 | −0.5 | 0.951 | 0.264 | −0.392 | 0.048 | 0.889 | 0.374 | 0.348 |
| Sh2g | 182.0 | −3.9 | −0.5 | 0.377 | 0.394 | −0.352 | 0.044 | 0.924 | 0.392 | 0.696 |
| $H_O$ | 182.4 | −3.5 | – | −0.572 | 0.757 | −0.357 | 0.044 | 4.260 | 1.847 | 0.386 |
| Sh2p | 184.0 | −1.9 | 1.6 | 0.584 | 0.360 | −0.352 | 0.043 | 0.694 | 0.359 | 0.623 |
| Sh6e | 184.1 | −1.8 | 1.7 | 0.716 | 0.318 | −0.354 | 0.043 | 0.639 | 0.330 | 0.522 |
| Sh6L | 185.1 | −0.8 | 2.6 | 0.974 | 0.268 | −0.355 | 0.043 | 0.795 | 0.471 | 0.130 |
| Sha023 | 185.3 | −0.6 | 2.8 | 0.857 | 0.293 | −0.360 | 0.043 | 0.527 | 0.327 | 0.464 |
| Sha037 | 185.6 | −0.3 | 3.1 | 0.779 | 0.325 | −0.354 | 0.043 | 0.503 | 0.332 | 0.551 |
| Sha001 | 185.7 | −0.2 | 3.3 | 1.148 | 0.263 | −0.358 | 0.043 | −1.206 | 0.878 | 0.058 |
| Sh3a | 185.8 | −0.1 | 3.4 | 1.250 | 0.286 | −0.358 | 0.043 | −0.514 | 0.358 | 0.304 |
| Base[4] | 185.9 | – | 3.5 | 1.090 | 0.257 | −0.357 | 0.043 | – | – | – |
| Sha024 | 186.3 | 0.4 | 3.9 | 1.177 | 0.269 | −0.355 | 0.043 | −0.539 | 0.436 | 0.203 |
| Sha011 | 186.5 | 0.6 | 4.1 | 0.915 | 0.295 | −0.351 | 0.043 | 0.397 | 0.334 | 0.348 |
| Sh2b | 186.7 | 0.8 | 4.2 | 0.965 | 0.278 | −0.362 | 0.043 | 0.369 | 0.332 | 0.420 |
| Sh2i | 186.7 | 0.8 | 4.2 | 0.959 | 0.282 | −0.361 | 0.044 | 0.364 | 0.330 | 0.420 |
| Sha028 | 186.7 | 0.8 | 4.2 | 1.216 | 0.285 | −0.359 | 0.043 | −0.392 | 0.358 | 0.319 |
| Sha010 | 186.7 | 0.8 | 4.3 | 1.493 | 0.458 | −0.362 | 0.044 | −0.459 | 0.421 | 0.812 |
| Sha025 | 186.9 | 1.0 | 4.5 | 1.013 | 0.268 | −0.360 | 0.043 | 0.365 | 0.371 | 0.261 |
| Sha015 | 187.2 | 1.3 | 4.8 | 0.990 | 0.282 | −0.362 | 0.044 | 0.274 | 0.330 | 0.493 |
| Sha012 | 187.5 | 1.6 | 5.0 | 0.994 | 0.294 | −0.359 | 0.043 | 0.214 | 0.324 | 0.507 |
| Sha026 | 187.6 | 1.7 | 5.2 | 1.050 | 0.267 | −0.360 | 0.043 | 0.194 | 0.362 | 0.304 |
| Sha008 | 187.6 | 1.7 | 5.2 | 0.990 | 0.317 | −0.358 | 0.043 | 0.176 | 0.332 | 0.609 |
| Sha042 | 187.7 | 1.8 | 5.2 | 1.140 | 0.280 | −0.356 | 0.043 | −0.161 | 0.343 | 0.362 |
| Sha033 | 187.7 | 1.8 | 5.3 | 1.181 | 0.339 | −0.362 | 0.045 | −0.143 | 0.338 | 0.391 |
| Sh5c | 187.8 | 1.9 | 5.4 | 1.077 | 0.262 | −0.356 | 0.043 | 0.168 | 0.675 | 0.058 |
| Sha014 | 187.9 | 2.0 | 5.5 | 1.104 | 0.293 | −0.356 | 0.043 | −0.034 | 0.325 | 0.536 |
| Sha034 | 187.9 | 2.0 | 5.5 | 1.082 | 0.270 | −0.356 | 0.043 | 0.036 | 0.417 | 0.174 |
| Sha009 | 187.9 | 2.0 | 5.5 | 1.081 | 0.309 | −0.356 | 0.043 | 0.017 | 0.327 | 0.420 |
| Sha036 | 187.9 | 2.0 | 5.5 | 1.096 | 0.300 | −0.357 | 0.045 | −0.017 | 0.388 | 0.203 |
| Sh2v | 187.9 | 2.0 | 5.5 | 1.086 | 0.298 | −0.357 | 0.043 | 0.009 | 0.323 | 0.493 |

**Notes.**
[1] Difference in AIC between the focal model and the "base" model, which excludes genetic data.
[2] Difference in AIC between the focal model and the multilocus heterozygosity model.
[3] Observed rate of heterozygosity in the sample set, for the specified locus.
[4] Sh2L was monomorphic in this subset of devils; the model is therefore excluded from the table, as it is identical to the "base" model.

Devil populations, with and without DFTD, are fragmented across the landscape, so inbreeding depression may be occurring at other sites, particularly those affected by DFTD. It would be informative to continue to quantify inbreeding depression into the future to facilitate effective management of wild populations. Evidence of inter-individual variation
in inbreeding at Woolnorth ($g_2$ analysis) indicates that we have the molecular tools available to test for inbreeding depression; the next step is to determine whether this is also true for other sites. Our results presented here contribute to the growing body of literature that is assisting the STDP to predict the outcomes of their management strategy of augmenting small wild populations to promote gene flow (*Fox & Seddon, 2019*; *Grueber et al., 2019*).

Extensive modelling will be informative for predicting the long-term consequences of reduced reproductive output on devil population dynamics and growth. At sites where DFTD is present, there have been observed increases in precocial breeding (1 year olds breeding) and increased litter sizes, highlighting the complex interplay between reproductive parameters and population sustainability in the presence of DFTD (*Jones et al., 2008*; *Grueber et al., 2018*; *Lazenby et al., 2018*). However, the degree to which this population compensation permits long-term population growth is still unclear (*Lazenby et al., 2018*), and it is also unclear whether similar processes will occur in the face of inbreeding depression. Previous modelling to assess the retention of rare alleles at DFTD-present sites when population sizes are small showed that population supplementation would be required to ensure long-term genetic viability (*Grueber et al., 2019*). To predict the long-term consequences of the inbreeding depression observed here, a more comprehensive modelling exercise is required. These models need to account for possible trade-offs and interactions among inbreeding, reproductive dynamics, changes in survivorship in the presence of DFTD, and other ecological parameters such as the impact of drought/changing climate, roadkill, habitat fragmentation etc. We also acknowledge that translocations carry risks, which are incorporated into management planning when determining the cost/benefit trade-off of supplementing wild populations (*Ewen et al., 2012*). For example, in addition to common concerns such as survival rates (e.g., *Thalmann et al., 2016*), the Save the Tasmanian Devil Program also considers the potential for vehicle strike (e.g., *Grueber et al., 2017*) and other behavioural factors (e.g., *Sinn et al., 2014*), microbiome changes (e.g., *Chong et al., 2019*) and variation in the genetic contributions of translocated individuals (e.g., *McLennan et al., 2018*). Consideration of the genetic impacts of translocation is critical for ensuring the long-term persistence of managed populations (*Weeks et al., 2011*). For wild devils, it would be valuable to consider the interplay between the apparent costs of inbreeding depression (this study) and genetic diversity loss (e.g., *Grueber et al., 2019*) in an analysis that also incorporates the effects of DFTD on demography (e.g., *Jones et al., 2008*; *Grueber et al., 2018*; *Lazenby et al., 2018*) and genetic structure (e.g., *Lachish et al., 2011*; *Epstein et al., 2016*).

## CONCLUSIONS

We have documented the first evidence of inbreeding depression in a wild population of Tasmanian devils. Whether inbreeding is the driver of the observed reproductive decline at Woolnorth, and/or whether the reproductive decline is driving an increase in inbreeding cannot be specifically determined. Although the long-term impact of this reduced productivity on population growth is unknown at this time, our data do show that inbreeding is detrimental to reproductive output in this population, and has the potential

to become more prevalent. Augmenting this population with genetic material from other locations across Tasmania may alleviate the effects of future inbreeding and minimise the occurrence of inbreeding depression. Ongoing monitoring after augmentation will provide valuable insights to the impacts of supplementation on population growth and inbreeding.

## ACKNOWLEDGEMENTS

We thank the Save the Tasmanian Devil Program for the collection of samples over the years of this study; this research could not be conducted without their hard work. Thanks also to B Lazenby for valuable comments on the draft manuscript and E Johnson for assistance with DNA extractions.

### Funding

This research was supported by funding from the Australian Research Council (grant number LP140100508), the Save the Tasmanian Devil Program, the Zoo and Aquarium Association Australasia and San Diego Zoo Global. The funders had no role in study design, data collection and analysis, decision to publish, or preparation of the manuscript.

### Grant Disclosures

The following grant information was disclosed by the authors:
Australian Research Council: LP140100508.
The Save the Tasmanian Devil Program.
The Zoo and Aquarium Association Australasia and San Diego Zoo Global.

### Competing Interests

Samantha Fox and David Pemberton are employed by Save the Tasmanian Devil Program. Catherine E. Grueber was also affiliated with San Diego Zoo Global at the time this work was conducted. Samantha Fox is an Adjunct Biologist to Toledo Zoo.

### Author Contributions

- Rebecca M. Gooley performed the experiments, analyzed the data, prepared figures and/or tables, authored or reviewed drafts of the paper, and approved the final draft.
- Carolyn J. Hogg conceived and designed the experiments, performed the experiments, analyzed the data, authored or reviewed drafts of the paper, and approved the final draft.
- Samantha Fox, David Pemberton and Katherine Belov conceived and designed the experiments, authored or reviewed drafts of the paper, and approved the final draft.
- Catherine E. Grueber conceived and designed the experiments, analyzed the data, prepared figures and/or tables, authored or reviewed drafts of the paper, and approved the final draft.

### Animal Ethics

The following information was supplied relating to ethical approvals (i.e., approving body and any reference numbers):

Samples were collected by the Save the Tasmanian Devil Program following their "Standard Operating Procedure" (see Appendix 5 in *Hogg et al., 2019* "Saving the Tasmanian Devil: Recovery Through Science-Based Management" CSIRO Publishing pp 301-303).

## Data Availability

Raw data is available in the Supplemental Files.

## Supplemental Information

Supplemental information for this article can be found online at http://dx.doi.org/10.7717/peerj.9220#supplemental-information.

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
