# Peer review of "Inbreeding depression in one of the last DFTD-free wild populations of Tasmanian devils"

_PeerJ, doi:10.7717/peerj.9220_

## Round 0.1 · original submission · Major Revisions

We have received three reviews on your manuscript. All reviewers found positive aspects about the study, but also raised a number of important points that deserve further revisions.

Importantly, all of them questioned the validity of the variable used (in particular the litter size estimates), of the statistical modelling approach, and thus the strength of the findings. Also, other indices should be added to assess temporal stability of inbreeding.

The revisions will need to integrate all detailed comments made by reviewers. Furthermore, I would like to see a locus by locus IR analysis, or at least a jackknifing over loci to assess if the results are due to some specific loci (local effect). Like one of the reviewer I would also suggest to tone down the inferences made from the results. These were obtained using neutral markers that are not always representative of the genome-wide inbreeding - this should be recognized in the discussion.

Reviewer 1 ·

Basic reporting

The manuscript was clearly written and the figures were well labeled. However, I had some trouble understanding some of the statistics relating to regression of IR to fitness (lines 109-115). I think if you provided the regression equation in the text, this would clear things up.

Experimental design

Overall, the design of the work is solid, but I think there is some opportunity for expanding the analyses. Were all 168 devils that were genotyped (line 84) adults? If not, it would be very interesting to identify parent offspring pairs each year, as this would provide an estimate of male reproductive output. You could even try to understand the accuracy of the estimate by comparing the field estimates of female reproductive output to estimates made from a genetic parentage analysis. In the same line of thinking, even with all adult samples you could examine your data for parent-offspring pairs across years and compare IR between parents with and without surviving offspring.

Validity of the findings

Although the authors seem to have verified their genotyping well (repeated genotyping of the same individual, use of microchecker), it is difficult for me to assess the validity of the regression results provided in table 2. At a minimum, I’d expect to see R2 estimates included in this table.

In addition to this, the comparisons over years is confusing. In the text, the results are a non-significant slope (line 122) between IR and year and Table 2 displays a significant relationship between year and the number of joeys. How much of the marginally significant relationship between IR and number of joeys is due to the effect of year (figure 2)? This should really be statistically addressed by using interactions in your statistical model, random effects, or some other method.

Finally, I am curious why a single genetic statistic – internal relatedness – was used in this study. I would like to see this potentially interesting result built out a little to really sell this as being the whole picture of this population. For example, what about other estimates of inbreeding (Fis, Queller and Goodnight’s R), do they also relate to fitness? What were the drivers of the year-to-year differences in fitness (if abiotic data can be obtained, for example)? These and many other analyses could make this paper much more substantial.

Additional comments

Overall, I think this manuscript has merit, provided the authors clarify their statistical approach, and shore-up the issues noted above. In addition to this, I caution the authors on overselling their inferences. For example, the first paragraph of the discussion includes statements suggesting that devils may be “close to the tipping point” of the inbreeding depression spiral, despite the marginal significance noted in figure 2. This paragraph, and really the intense tone of the entire paper, is somewhat contradictory to the strength of the findings.

Reviewer 2 ·

Basic reporting

no comment

Experimental design

see general comments to the author

Validity of the findings

see general comments to the author

Additional comments

The introduction sets up the paper by suggesting that inbreeding depression can affect population persistence (which it can in some cases). The paper then tests for an association between heterozygosity and an individual fitness component (number of offspring born to a female). However, it is not discussed whether this fitness component strongly affects population growth in devils; it is thus not known whether the observed inbreeding depression detected is important to population persistence. This issue is relevant to much of the Discussion also. I suggest trying to address this with methods borrowed from population ecology (i.e., a sensitivity analysis) if at all possible; this would really be a great addition to devil population dynamics and management.

82: Is there good evidence that the count of active teats is actually a reliable measure of the number of offspring? Can a single offspring result in multiple active teats? What proportion of reproduction estimates are based on the count of active teats?

It is said that the microsatellites are neutral (line 89), and this may well be true. However, I assume it is unknown whether each of these loci is closely linked to loci that are responsible for some of the variance in fitness, in which case their evolution would be affected by linked selection. I suggest just saying that microsatellites were used, and that the analyses assume that they are unaffected by natural selection.

121-122: One of the findings in this study is that IR remained temporally constant. This finding could be sensitive to how the allele frequencies were calculated, so I suggest clarifying this. What year classes of samples were used to calculate IR for all of the individuals? If IR was calculated based on allele frequencies estimated for each year class of individuals separately, then IR may remain roughly constant through time even if heterozygosity has been lost through time. I suggest showing a plot of multiple-locus heterozygosity versus year (which does not account for allele frequencies), and testing if this showed a temporal decline in the study population.

164-166: You would really need to quantify the effect of the observed inbreeding depression for litter size on population growth rate to make this useful for population management. It seems like the data on vital rates needed for a sensitivity analysis (to assess the importance of litter size to population growth) may well available for devils given the intensity of the monitoring program.

Reviewer 3 ·

Basic reporting

No comment.

Experimental design

I made a few suggestions regarding the analysis for the authors to consider.

I think more explanation and detail should be provided regarding how the litter count data were obtained in the field.

Validity of the findings

No comment.

Additional comments

Gooley et al. present a straightforward study investigating inbreeding effects on reproductive data from Tasmanian devils. In general the study is well done and the paper is well written. I especially appreciated the concise manner in which the paper is written. I had several comments for the authors to consider that, if addressed, might help with clarity of methods and might provide additional information from the data and analysis. They are listed in order of importance.

1. Line 109-113: Using binomial regression in this fashion seems like a reasonable approach. However, I think there might be value to considering probability of parturition in a given year and litter size separately. For probability of giving birth you could use a binomial response of 0 = did not give birth and 1 = gave birth to at least 1 offspring. For litter size, you could cumulative logit regression for the devils that did produce young (response 1-4). See Hostetler et al. 2012, Oecologia 168:289-300 for an example of a study addressing very similar questions as yours with inbred and outbred Florida panthers. They first modeled probability of giving birth and then modeled litter sizes using the regression approaches I mentioned.

My reasoning for this suggestion is you have many, many 0’s in your dataset as evidenced by Fig. 2. Do you know anything about the nature of those 0’s? It seems like what you have modeled might not actually be litter size but rather a combination of probability of breeding/parturition and litter size. By modeling these separately you would know whether inbreeding influences 1) the probability of giving birth at all and 2) litter size. Given all those 0’s and the fact that it looks like most females that actually had young had 3 or 4 offspring, it may be that inbreeding is actually affecting the probability of giving birth rather than litter size per se.

2. Lines 79-84: Can you be a bit clearer and provide more detail about the litter size evaluation methods? It sounds like either you 1) observed young in the pouch or 2) counted the number of active teats. Can you explain the timing of these young and teat observations relative to the birth of the offspring? Is there any chance some young had died prior to you observation of the actual offspring in the pouch? Do the number of active teats accurately reflect litter size and is this unambiguous to assess in the field? How long can an active teat be detected after the young are born?

3. 109-115: Were each of your 90 datapoints for the regression analysis litter counts for different individuals or do you have multiple litter counts from some females? If that latter, those data would obviously not be independent and would need to be accounted for with a random effect of individual.

4. Line 125-128: I think it would be worth doing a simple model selection process (e.g., using AIC) to see which of the variables are retained in the model or models with the strongest support. You appear to have strong effects of year and inbreeding but it would be good to see what the best models were.

5. Line 96-100: Did you also estimate the other individual-level heterozgosity metrics that can be estimated using Rhh? Were these also strong predictors of reproductive values?

6. Line 40: I suggest revising and removing the word “rallied”. I suggest simply explaining the research and conservation actions that have been undertaken.

---

## Round 0.2 · Minor Revisions

We have received two reviews on your revised version (from previous reviewers). Both were pleased with the modifications performed but still raised a number of important additional points that should be addressed. In particular, reviewer 1 asked for further clarifications of the context of the study in the Introduction and regarding the year effect in the analyses. Reviewer 2 suggested an additional way to separate local vs global effects and asked for a more thorough assessment of whether inbreeding depression is likely to affect population dynamics here.

Reviewer 1 ·

Basic reporting

I wonder if something is missing from the introduction. Perhaps this is obvious to the authors, but it took me until the middle of the second round of reviewing to think that maybe the reason this population does not have DFTD is because of its extreme isolation. If this is indeed the case, I suggest the authors put this information up front in the introduction, as it really does speak to the likelihood of increasing severity of inbreeding depression. To me, this is a much more compelling line of thought than the wild vs. captive inbreeding depression effects currently put forth in the text.

Line 33: “in areas infected by an infectious clonal cancer” the 2 x infect* is a little confusing to read. Please consider rephrasing.

Line 103: Should be ‘Sambrook’ instead of ‘Sanbrook’

Experimental design

The questions are clearly stated and analyses conduct are robust. One minor thing is that I do not understand the format used for providing the regression equations. Shouldn’t there be coefficients with subscripts to denote individuals and error terms in the equations?

Secondly, and perhaps more importantly, I am still a little unsure about how to separate the year effects from the IR effects (displayed in Table 2). The year-to-year effects seem much stronger than the IR effects, which is troubling because the authors did not see increasing inbreeding over time (lines 203-204). How much do the environmental effects influence fitness? Including year as a random effect in the model may allow the authors to tease these the IR and environment effects apart.

Validity of the findings

Given the relatively weak relationship between IR and fitness, I am not sure it makes much sense to supplement the population, as that is not without risk. I suggest this be toned down in the paper.

Additional comments

I am overall happy with the revisions the authors have made to their manuscript.

Reviewer 2 ·

Basic reporting

See attached PDF

Experimental design

See attached PDF

Validity of the findings

See attached PDF

Additional comments

See attached PDF

Annotated reviews are not available for download in order to protect the identity of reviewers who chose to remain anonymous.

---

## Round 0.3 · Minor Revisions

I am generally pleased with the answers provided by the authors.
However, I think the analysis including year as a random effect (previously suggested by one of the reviewer) should be conducted. Both year as a random factor (to control for non-independence of data but also for environmental differences among years) and year as a continuous variable (to control for the decline in fitness over time) are relevant and should be included in the same analysis. Contrary to the author claim, fitting year as a random factor would provide information on the importance of year-to-year variation (as provided by the amount of variance explained by the random component). When doing so, please provide both marginal and conditional R-squared.

Other minor points:
-include null allele proportions for each locus in table 1.
-clarify in methods if different individuals could be identified among years (possible pseudoreplication).
-provide you alternative models for predicting reproductive success (with appropriate stats: aic, weight, etc.) as a supplementary material file.

---

## Round 0.4 · accepted · Accept

I am generally pleased by the additional modifications performed.

On L137-138 : Please use the wording ‘random effect’ instead of ‘random fixed factor’, which could be misleading.